Resource

# Single-cell RNA sequencing of human breast tumour-infiltrating immune cells reveals a γδ T-cell subtype associated with good clinical outcome

Katerina Boufea[1], Victor Gonzalez-Huici[1] , Marcus Lindberg[1] , Nelly N Olova[2], Stefan Symeonides[3] , Olga Oikonomidou[3], Nizar N Batada[1] 

**The association of increased levels of tumour-infiltrating gamma-delta (γδ) T cells with favorable prognosis across many cancer types and their ability to recognize stress antigens in an MHC unrestricted manner has led to an increased interest in exploiting them for cancer immunotherapy. We performed single-cell RNA sequencing (scRNA-seq) of peripheral blood γδ T cells from healthy adult donors and from fresh tumour biopsies of breast cancer patients. We identified five γδ T cells subtypes in blood and three subtypes of γδ T cells in breast tumour. These subtypes differed in the expression of genes contributing to effector functions such as antigen presentation, cytotoxicity, and IL17A and IFNγ production. Compared with the blood γδ T cells, the breast tumour-infiltrating γδ T cells were more activated, expressed higher levels of cytotoxic genes, yet were immunosuppressed. One subtype in the breast tumour that was IFNγ-positive had no obvious similarity to any of the subtypes observed in the blood γδ T cell and was the only subtype associated with improved overall survival of breast cancer patients. Taken together, our study has identified markers of subtypes of human blood γδ T cells and uncovered a tumour-infiltrating γδ T cells subtype associated improved overall cancer survival.**

## Introduction

Whereas the conventional population of T cells use a CD3-associated alpha/beta ($\alpha\beta$) TCR for recognition of processed peptide antigens presented on MHCs, a minor population of T cells express a gamma-delta ($\gamma\delta$) TCR, which can recognize both peptide and non-peptide antigens directly (Sebestyen et al, 2020). In human peripheral blood, two major γδ T-cell subsets are defined based on the type of variable segment of the delta (V$\delta$) chain that is present in their TCR and there is non-random pairing between V$\delta$ chains and Vγ chains. For instance, the Vγ9-V$\delta$2 γδ T-cell subtype, which carries out response to transformed cells and invasive pathogens, makes up between 50% and 90% of the γδ T cells in the human peripheral blood (Dimova et al, 2015).

In addition to the TCR composition–based classification, γδ T-cell subtypes have also been distinguished based on their effector functions. Two broad subgroups defined based on cellular function include effector γδ T cells, which can kill cells such as tumours directly (Kabelitz et al, 2007), and regulatory γδ T cells, that promote immunity through secreting cytokines (Zhao et al, 2018). For example, mouse γδ T-cell have a functionally well-defined and mutually exclusive IL17A producing and an IFNγ-producing subtypes (Ribot et al, 2009).

Using a candidate gene approach, Ryan et al (2016) have identified *CD16* and *CD28* as markers of two distinct blood V$\delta$2 subtypes. However, subtypes and markers of V$\delta$1 have remained unclear. Pizzolato et al (2019) carried out single-cell RNA sequencing (scRNA-seq) of V$\delta$1 and V$\delta$2 sorted subsets of human blood γδ T cells. Although their data allowed identification of genes differentially expressed between the sorted V$\delta$1 and V$\delta$2 subsets of γδ T cells, the relatively low number of cells that were sequenced did not allow identification of subclusters within them. Additional deep scRNA-seq provides an opportunity to uncover γδ T subtypes, their markers, and putative functions unbiasedly (Regev et al, 2017).

Whereas the human γδ T cells have been characterized largely in the peripheral blood and in the context of bacterial and viral infections, computational study of The Cancer Genome Atlas (TCGA) cancer data has shown that elevated levels of γδ T cells in a variety of solid tumours are associated with favourable prognosis (Ma et al, 2012; Gentles et al, 2015; Wu et al, 2019). Using ex vivo grid culture expansion of γδ T cells isolated from breast tumours, Wu et al (2019) identified an IFNγ-positive innate-like δ1 subtype that was associated with favorable overall survival in triple-negative breast

---

[1]Centre for Genomic and Experimental Medicine, University of Edinburgh, Edinburgh, Scotland   [2]MRC Human Genetics Unit, University of Edinburgh, Edinburgh, Scotland   [3]Cancer Research UK Edinburgh Centre, University of Edinburgh, Western General Hospital, Edinburgh, Scotland

Correspondence: nizar.batada@gmail.com
VíctorGonzález-Huici's present address is Institute for Research in Biomedicine, Parc Científic de Barcelona, Barcelona, Spain

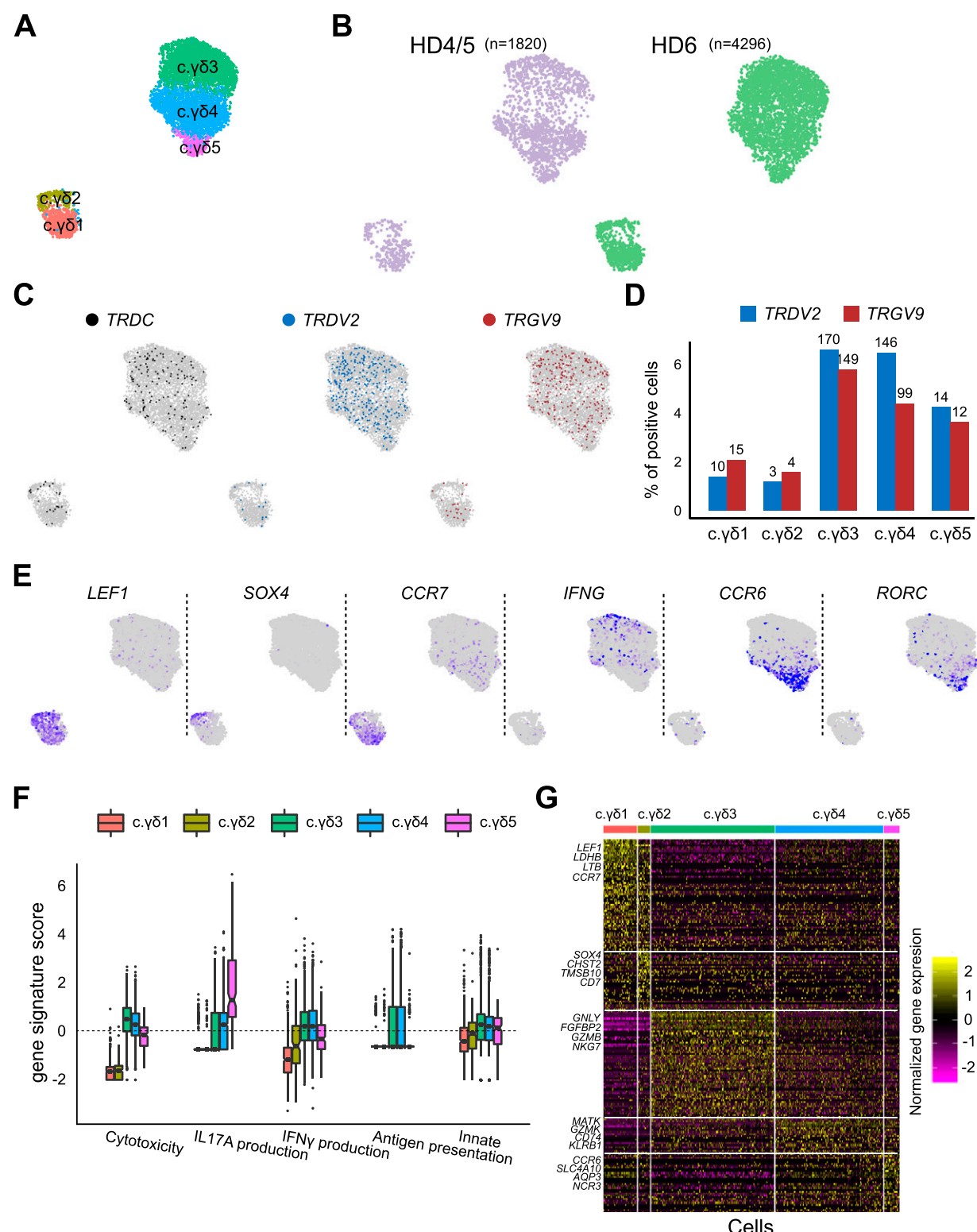

**Figure 1. Unsupervised analysis of single-cell RNA sequencing (scRNA-seq) data on γδ T cells from peripheral blood of healthy adult donors identifies multiple subtypes.**
**(A)** UMAP-based projection of the merged single cell gene expression data of blood derived γδ T cells from three healthy donors. Different clusters are named arbitrarily with c.γδ prefix (to indicate γδ cells from circulation). **(B)** Overlay of data source on UMAP-based projection of scRNA-seq data. Cells from donor HD4 and HD5 were pooled before performing scRNA-seq and labeled as HD4/5. The number of cells from each dataset is shown above the projection. **(C)** Labeling of cells positive for TCR delta genes. Overlay of cells that have genes mapping to the *TRDC* (left), *TRDV2* (middle) and *TRGV9* (right) gene segments. **(D)** Quantification of *TRDV2* and *TRGV9* in each cluster.

cancer patients. However, markers unique to this subtype and the gene expression programs that potentially underlie their clinical association have not been defined. Here, we generated a large reference-level scRNA-seq dataset on blood γδ T cells from three healthy donors and from two breast tumours. Our large dataset containing transcriptomes from a total of ~7,000 γδ T cells identified multiple novel subsets of and marker genes for both δ1 and δ2 subtypes, including a δ1 breast tumour-infiltrating γδ T-cell subtype that is absent in blood and is associated with favorable overall survival of breast cancer patients.

# Results

### Unsupervised clustering of scRNA-seq of blood γδ T cells

Peripheral blood γδ T cells from three healthy donors were sorted using anti-TCR γδ antibody (see the Materials and Methods section) and subjected to 10x Chromium (Zheng et al, 2017) single-cell 3′ gene expression library generation and subsequently sequenced on Illumina NovaSeq S1 platform. Cells from donors HD4 and HD5 were pooled before performing scRNA-seq and labeled as HD4/5. Raw sequence reads were processed using the Cell Ranger pipeline (see the Materials and Methods section). The resulting gene expression counts for each sample were merged using Seurat (Butler et al, 2018; Stuart et al, 2019). Only clusters that were *CD3E* and TCR δ constant gene segment (*TRDC*), which is a pan-γδ T-cell maker, positive and present in both the datasets were retained and the final combined data comprised 6,116 cells with an average of 1,084 genes per cell representing 12,943 different genes in total.

Community based clustering (as implemented in Seurat) identified five clusters and two-dimensional projection of the merged scRNA-seq data using UMAP (McInnes et al, 2018 *Preprint*) revealed two macroclusters (Fig 1A). Clustering seemed not to be affected by confounding technical or biological variation: (1) all clusters had similar proportion of cells in each phase of the cell cycle suggesting that the clusters are not driven by cell-cycle differences, (2) removing cell-cycle genes did not affect clustering, and (3) clustering of subsamples of the data had consistent clustering (Fig S1). Moreover, clustering of datasets separately (HD4/5 or HD6) had the same number of clusters as that seen in the merged data and, in the merged data, cells from each of the two datasets (HD4/5 and HD6) were relatively evenly distributed among the clusters (Fig 1B). Only the *TRDV2* (TCR Vδ2 variable gene segment) and *TRGV9* (TCR Vγ9 variable gene segment) transcript reads were the only ones of the TRD/TRG gene family that were unambiguously mappable with the short (90 base pair) transcript read sequences that comprised our data (see the Materials and Methods section; Fig 1C and D).

To assess if the clustering is biologically meaningful, we looked at the distribution of various known markers and curated gene signatures that associate with effector functions known to be carried out by γδ–T cells (Fig 1E–G). As expected, we observed mutual exclusive expression of markers defining the IFNγ and IL17A producing subtypes that define two well-characterized γδ–T clusters. Furthermore, the cluster (c.γδ5) differentially expressed *CCR6* and RORC, which are well known markers of IL17A producing cells (Haas et al, 2009). Based on the above data, we suggest the following functional annotations for these clusters: (i) the c.γδ1 cluster appears to be naive or immature (as indicated by *LEF1*), (ii) the c.γδ2 cluster appears to be recent thymus emigrants (as indicated by *CCR7*) (Reinhardt et al, 2014), (iii) the c.γδ3 and c.γδ4 clusters appears to carry out canonical Vδ2 cytotoxic functions namely antibacterial, antigen presentation function and IFNγ producer; and (iv) the c.γδ5 cluster is a *IL17A* producer (which was enriched for *RORC* and *CCR6* expression) (Singh et al, 2008).

Ryan et al (2016) identified two different anti-*TRDV2+* subtypes in blood γδ T cells that are marked by mutually exclusive expression of *CD28* and *CD16*, which we also observe in our data (Fig 2A). Whereas *CD16* was specifically enriched in only one IFNγ+ cluster (i.e., c.γδ3) that is also *TRDV2*^high (Fig 1C and D), *CD28* was more diffused and present in all the other subtypes including the smaller macrocluster that had low levels of *TRDV2* (Fig 1C and D). To corroborate this with a larger gene set, we scored all the subclusters for the published gene signatures of the γδ-T-cell partitioning defined by *CD16* and *CD28* (Ryan et al, 2016). As before, whereas *CD16* gene signature was exclusive to one subtype of γδ T cells, the *CD28* gene signature was enriched in all the other subtypes, including the two of the smaller macrocluster subtypes (Fig 2B). Identification of cluster enriched genes via unsupervised clustering (Fig 1G) identified *CXCR6*, a surface protein for which commercial antibody existed, which suggested that is may be a better marker than *CD28* for isolating *CD16* negative Vδ2 cells, because unlike *CD28*, it is absent in the smaller macrocluster that is low in *TRDV2*. To test this hypothesis, we performed immunostaining with anti-*CXCR6* (present in both c.γδ4 and c.γδ5) and with anti-*GPR56* (present in c.γδ3) (Fig 2C). We stained blood from three donors (including two additional healthy donors, HD9 and HD10) with anti-CD3, anti-TCRγδ–Vδ2, anti-GPR56, and anti-CXCR6 (Fig 2D). The L-shaped scatter plot of *GPR56* and *CXCR6* suggests mutual exclusion and corroborates they mark different δ2 subpopulations within human blood γδ-T population. Thus, the clusters identified in our data are consistent with what has been reported in the literature regarding human blood γδ T cells and our analysis allowed us to identify a better marker for separating these subtypes.

### Identification of γδ T-cell subtypes within breast tumour microenvironment

To uncover γδ T-cell subtypes and their gene expression programs in the tumour microenvironment, we dissociated fresh triple-negative (TNBC) and Her2+ breast tumour biopsies into single

---

y-axis shows the per cent of cells within each cluster (x-axis) of the merged data that is positive for *TRDV2* (blue) and *TRGV9* (red) gene segments. Numbers above the bars show the number of positive cells. **(E)** UMAP of PBMC cells coloured by the expression of a selected set of markers, *LEF1, SOX4, CCR7, IFNG, CCR6*, and *RORC*. Grey indicates zero expression and purple indicates high expression. **(F)** Scores of curated effector gene sets for IFNγ production, IL17A production, cytotoxicity, adaptive (i.e., antigen presentation on MHC class 1), and innate gene sets in each of the blood γδ T-cell cluster (x-axis). **(G)** Heat map showing genes (rows) enriched in each of the cluster (columns). Yellow represents enrichment and purple represents depletion. Only the top four genes per cluster are labeled.

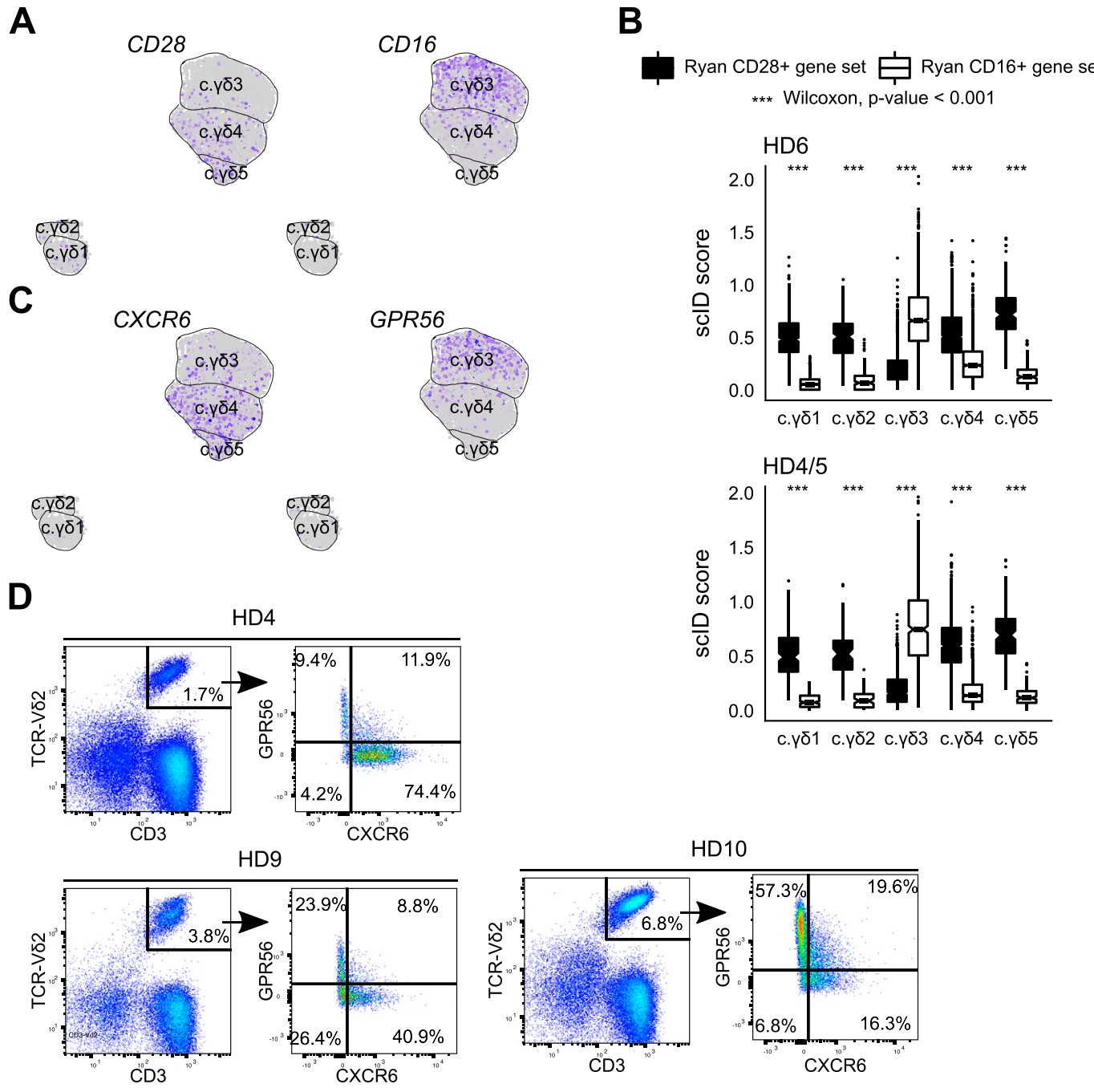

**Figure 2. Validation of the blood γδ T-cell subtypes.**
**(A)** Feature plot showing *CD16* and *CD28*, which are published markers of the δ2 subtype of blood γδ T cells (Ryan et al, 2016). Each dot is a cell. Grey indicates no expression and purple indicates high expression. **(B)** Scores for published gene signatures of *CD16* (white) and *CD28* (black) δ2 subtypes in the clusters found in our blood γδ T-cell scRNA-seq data (x-axis). *P*-values were computed using Wilcoxon signed-rank test. **(C)** Feature plot showing expression of GPR56 and CXCR6, markers that appear to be mutually exclusive in blood δ2 subtypes. **(D)** Flow cytometry based validation of novel markers, *GPR56* (y-axis) and *CXCR6* (x-axis), in peripheral blood γδ T cells TCRδ2 subtypes. Healthy donor identities (which include blood γδ T cells from two new donors, HD9 and HD10) are indicated in the title. Numbers in each quadrant indicate percentage of δ2 cells.

cells and performed scRNA-seq on all immune cells (*CD45*⁺) (see the Materials and Methods section). Three clusters that were double positive for *CD3* and *TRDC* were present in both the BC samples and thus designated as γδ T cells (Fig 3A and B). All the three clusters were relatively uniform in TRDC but only one cluster (γδ-T.3) appeared to be *TRDV2*+ (Fig 3C and D).

To identify *IL17A+* and *IFNγ+* subtypes, we looked at the expression of their markers in the γδ-T clusters indicating that cluster

2 is an *IFNγ* producer and cluster 3 is *IL17A* producer (Fig 3E). Supervised scoring of clusters with genes defining effector functions, as performed previously, indicated that γδ-T.1 is highly cytotoxic, γδ-T.2 was most innate-like subtype and also had the highest levels of IFNγ production and antigen presentation. γδ-T.3 was the only IL17A producing subtype (Fig 3F).

We next investigated if these γδ T-cell BC clusters had different clinical impact. Using mathematical deconvolution to infer proportion of various immune cell types from bulk RNA-seq data, elevated levels of total γδ T cells were found to be associated with better survival in breast cancer (Ma et al, 2012) and across all cancers (Gentles et al, 2015); however, recent work observed an improved overall survival in TNBC breast cancer for a Vδ1 innate-like subtype of γδ T cells but not for the overall levels of γδ T cells (Wu et al, 2019). We sought to repeat this analysis but using BC γδ T-cell subtype–specific gene signatures identified here. We first confirmed that the gene signatures of each cluster were indeed specific by scoring each of the CD45⁺ immune clusters in BC1 for the three BC γδ-T signatures. The result suggested that these gene signatures were highly specific for the respective BC γδ T-cell clusters (Fig S2). Survival analysis of the breast cancer data from TCGA (Ciriello et al, 2015), indicated that only the enrichment of BC γδ-T.2 gene signature, but not the signature of the other two BC γδ T-cell clusters, is associated with improved survival (Fig 3G). Surprisingly, the TCGA BRCA patients with higher scores of BC γδ-T.2 gene signature had lower levels of predictors of better overall survival i.e. expression of *αβ* T cell markers, cytolytic scores (Rooney et al, 2015) and tumour mutation burden compared to the TCGA BRCA patients that had lower scores of BC γδ-T.2 gene signature (Fig S3). *NCAM1*, a marker known to be very high in activated cells and not just in NK cells (Van Acker, Capsomidis et al, 2017), was the top most differentially expressed gene in this clinically relevant BC γδ T-cell subtype (Fig 3H).

### Comparison of blood and breast tumour-infiltrating γδ T cells

Last, we sought to compare and contrast breast tissue infiltrating γδ T cells with their counterparts in the peripheral blood. We first focused on selected genes relevant for anti-tumour immune function. Overall, the BC γδ T cells were more activated, had higher abundance of transcripts of genes involved in cytotoxicity and exhaustion, whereas the markers of memory and naive T cells were significantly lower in breast tumour γδ T cells compared to those in blood (Fig 4A). To identify equivalent clusters across BC and blood γδ-T datasets, we used scID (Boufea et al, 2020) which revealed that BC γδ-T.1 cluster was equivalent to blood c.γδ3 cluster (and consistent with both being marked by *CD16* and positive for IFNγ) and BC γδ-T.3 cluster was equivalent to blood c.γδ5 cluster (and consistent with both being marked by *CCR6* and positive for *IL17A*) (Figs 4B and C, 1E, and 3E). However, none of the blood clusters were similar to BC γδ-T.2 cluster, which was defined by expression of *NCAM1* (*CD56*) (Fig 4C). These results suggest a refined classification of human γδ T cells based on our scRNA-seq data: *SOX4* and *NCAM1* can distinguish the three *TRDV2*low clusters, whereas the combination of *CD16* and *CCR6* can be used to define the three *TRDV2*high subtypes (Fig 4D and Tables S2–S6).

## Discussion

Human γδ T cells are a poorly understood subset of T cells with untapped potential for exploiting them for immunotherapy. They are canonically defined either by the Vδ chains in their TCRs or by production of *IL17A* or *INFγ*. Unbiased characterization of their subtypes, markers that can be used for isolation of these subtypes and the gene expression programs that define these subtypes will provide a much needed resource for functional characterization and exploitation of these primitive cells that, unlike conventional T cells, can respond to both peptide and non-peptide stress antigens in both an adaptive or innate-like mode (Hayday, 2019).

To better characterize human γδ T-cell subtypes and uncover their gene signatures and markers, we performed scRNA-seq of pan γδ T cells from the human blood and breast tumour samples. We identified five subtypes of human blood γδ T cells, three of which were *TRDV2*high positive and two were *TRDV2*low (Fig 1C and D). In the human breast cancer setting we identified three subtypes, one of which was classified as *TRDV2*high while the TCR Vδ chain identity of the other cluster (γδ-T.1 and γδ-T.2) was unclear (Fig 3C and D). Thus, within the same TCR defined γδ T-cell compartment, there is heterogeneity in gene expression program and possibly effector functions. Two of the three breast tumour γδ T-cell subtypes, had a counter part in the blood γδ-T cells (Fig 4B); however, the one subtype that did not have a counter part in the blood was the only subtype associated with better overall survival in a large cohort of breast cancer patients characterized by the TCGA consortium (Ciriello et al, 2015). Thus, in agreement with another recent publication (Wu et al, 2019), only specific subsets of γδ T cells rather than the total γδ T cell level (Gentles et al, 2015) is associated with improved clinical outcome. Based on computational characterization of effector functions (Fig 3F), the clinically associated γδ-T.2 subtype appears to be similar or identical to the one identified recently using functional assays (Wu et al, 2019). We have identified a gene signature of this clinically associated subtype, which can be useful in their isolation and use as a potential prognostic biomarker of survival of breast cancer patients.

In the blood, the two *TRDV2*low subtypes can be distinguished based on their immaturity (*LEF1* and *CCR7*), whereas the other appears to be recent thymus emigrant that has a moderate cytotoxic and antigen potential (Fig 1E). The three blood *TRDV2*high subtypes can be distinguished based on their gene expression programs—the *CCR6* subtype is an *IL17A* producer and the other two subtypes (*CD16* and *CCR7⁻CD16⁻*) are IFNγ producers but the latter appears to have a higher capacity for antigen presentation and the former has higher cytotoxic potential (Fig 1G). Our data are consistent with the previously described *CD16/CD28* axis in the TCR-Vδ2+ compartment (Ryan et al, 2016). We identified and validated *GPR56* as an alternative marker to *CD16* and *CXCR6* as more specific marker of the two non-*CD16* *TRDV2*high subtypes (Fig 2).

In breast tumour, just like its counterpart in the blood (Fig 4B), the *CCR6* defines an *IL17A* producer and the *CD16* subtype is an *IFNγ* producer with high cytotoxic potential (Fig 3G). The breast tumour *NCAM1*-positive subtype that does not appear to have a counter part in the adult peripheral blood, is a strong *IFNγ* producer and higher expression of innate gene signature (Fig 3F). We suggest that

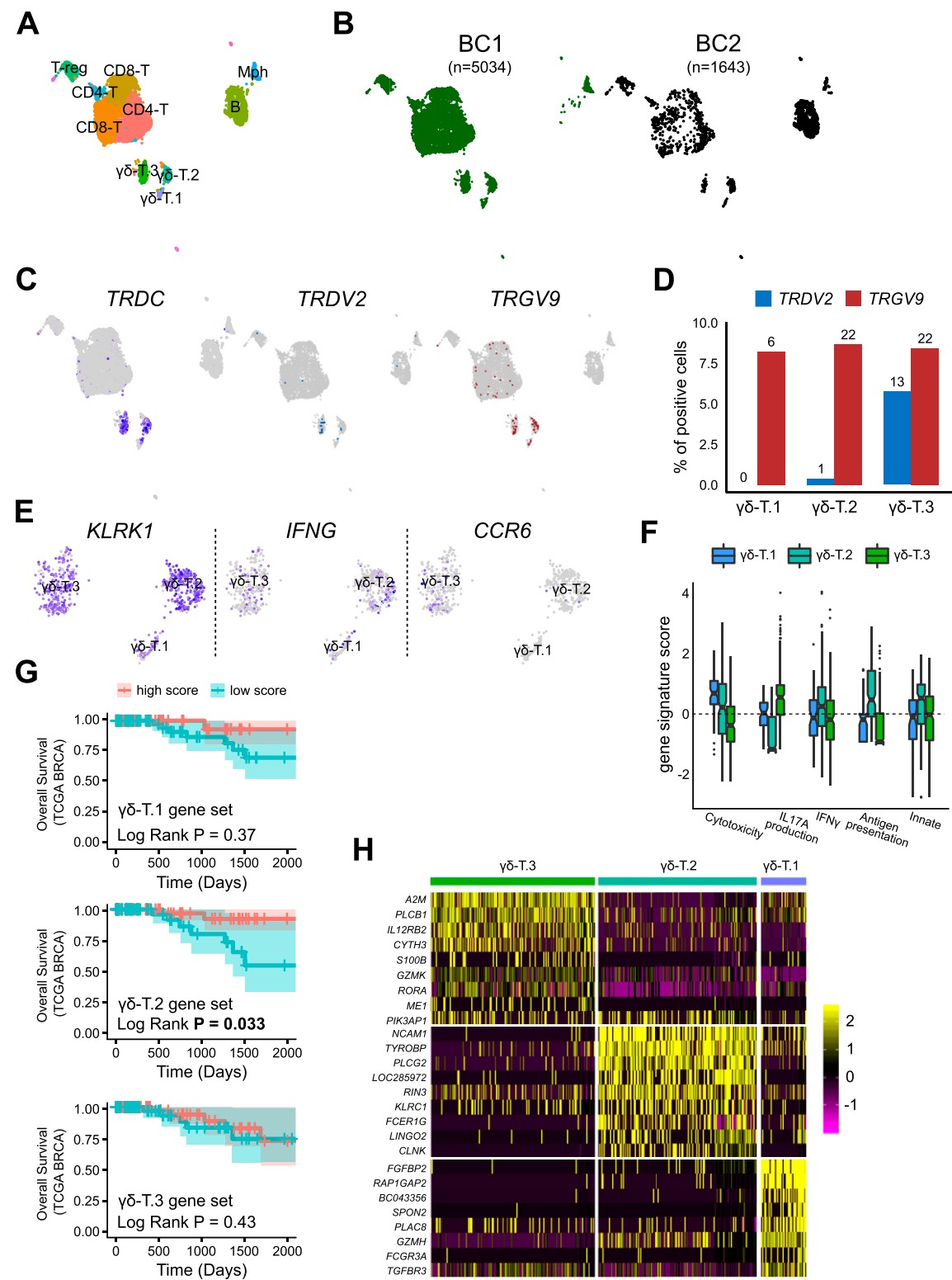

**Figure 3. Characterization of breast tumour-infiltrating γδ T cells uncovers a subtype that is associated with favourable outcome.**
**(A)** UMAP-based projection of the merged single-cell gene expression data of breast tumour-infiltrating immune cell datasets from two patients. Three clusters were double positive for *CD3* and *TRDC* were classified as γδ T cells (CD3E+). Mph, Macrophage (CD14⁺), T-reg (FOXP3+), regulatory T cells; B, B cells (CD19⁺); CD8-T, CD8⁺ αβ T cells; CD4-T, CD4⁺ αβ T cells. **(B)** Overlay of donor identity on UMAP-based projection of scRNA-seq data. The number of cells from each donor is shown above the projection. BC1 is a triple-negative subtype and BC2 is a Her2+ subtype of breast cancer (BC). **(C)** Identification of cells positive for genes encoding TCR δ chain. Overlay of cells that have genes mapping to the *TRDC* (left) and *TRDV2* (right) gene segments. **(D)** Quantification of enrichment of genes encoding *TRDV2* and *TRGV9* γ chain in the BC γδ T

the *NCAM1+* subtype is likely equivalent to the one recently found by Wu et al (2019) who discovered that a Vδ1+ *IL17A*^neg*IFN*γ+ subtype was in association with better overall survival of triple-negative breast cancer patients. Our studies corroborate and complement each other as we reached this conclusion through an unbiased gene expression approach, while their study reached the same conclusion via TCR sequencing and functional assays.

In summary, we have generated a large single-cell RNA-seq dataset on γδ T cells and have uncovered subtypes, potential functions, and markers. Our study is significant because (1) it identified new markers and refined classification (Fig 4D) that will help in isolation of γδ T-cell subtypes for further functional characterization; (2) it identified previously hidden gene expression heterogeneity in both the *TRDV2*^low and *TRDV2*^high subtypes and (3) it identified gene signature and a marker of a *IFNG+ TRDV2*^low subtype in breast tumours that is associated with better survival of breast cancer patients. Our analysis has contributed to a better understanding of the functional diversity of γδ T cells, and our data will serve as a valuable resource for the community that can be mined to identify markers useful for isolating novel subsets to be able to further interrogate their functions.

# Materials and Methods

## Sample acquisition, processing, and sorting

Breast cancer patient samples and blood from healthy donors were obtained with consent (NHS Lothian, Tissue Request No. 2017/SR865). Within ~30 min of obtaining the donor blood, blood was isolated via the Ficoll density gradient centrifugation following the manufacturer's protocol and cryopreserved in 10% DMSO. Cryopreserved blood were thawed rapidly and then incubated for 30 min at 4 degrees with appropriate antibodies (Table S1). Fresh breast tumour biopsies were obtained and processed within 1 h of surgical resection. Tumour tissue was first manually dissected and then chemically dissociated with a cocktail of Liberase DL/TL (Roche) for 30 min and then immunostained with anti-CD45 (Table S1) antibody. Dissociated cells were sorted for live singlet cells with appropriate gating strategy on BD FACSAria II (Fig S1).

## Single-cell sequencing

Sorted cells were counted on Countess and the gated population and the number of cells per sample that were loaded on Chromium 10× (version 2) were as follows: (1) HD4/5 (*CD45*^+*CD3*+pan-TCR γδ+; 10,000 cells from HD4 + 10,000 cells from HD5) and loaded on one lane; (2) HD6 (*CD45*^+*CD3*+pan-TCR γδ+; 20,000 cells were loaded on

one lane); (3) BC1 (*CD45*^+; 18,000 cells were loaded on one lane); (4) BC2 (*CD45*^+; 12,000 cells were loaded on one lane). scRNA library generation was carried out according to the manufacturer's guidelines. Libraries from HD4/5 and HD6 were sequenced on NovaSeq S1 with 26 + 90-bp reads. Libraries from BC1 and BC2 were sequenced on HiSeq2500 with 75 + 75-bp reads. Raw FASTQ format sequence reads were processed using the Cell Ranger pipeline for human genome hg19 assembly. Counts data from the cellranger count (Zheng et al, 2017) pipeline were further filtered if they were present in fewer than 0.2% total cells or 10 cells or had high levels of mitochondrial genes (>2%).

## Annotation of cells positive for the various TCR gamma and delta gene segments

To determine the identity of the TRD and TRG gene segments present in each cell, FASTQ reads from each cell were aligned to the VDJ gene sequences (refdata-cellranger-vdj-GRCh38-alts-ensembl-2.0.0 from Cell Ranger) using BWA. Reads uniquely mapping (mapping quality > 30) to *TRDV2* and *TRGV9* gene sequences were the only ones specifically enriched in the *TRDC* positive clusters in the CD45 positive cells in the BC1 data, thus were the only ones retained for further analysis (Supplemental Data 1).

## Generation of merged analysis-ready γδ T-cell scRNA-seq datasets

Post-filtered blood γδ T-cell scRNA-seq datasets (HD4/5 and HD6) were integrated and clustered using the Canonical Correlation Analysis (Butler et al, 2018) from the Seurat package (version 3.1.0) using the first 20 principal components. *TRDC* positive clusters from the merged dataset were retained. The final blood γδ T-cell data used here consisted of 6,116 cells. Post-filtered breast tumour scRNA-seq datasets (BC1 and BC2) were integrated and clustered using Canonical Correlation Analysis from the Seurat package using first 20 principal components. *TRDC* positive clusters from the merged dataset were retained for subsequent analysis.

## Identification of differentially expressed genes in single-cell RNA-seq datasets

The MAST function implemented in the Seurat package was used to identify differentially up-regulated genes between $\delta_1$ and $\delta_2$ subtypes. To identify tissue specific differences between γδ T cells from healthy blood and BRCA tissues, we converted each single-cell dataset into pseudo-bulk data, one per cluster, where the expression of each gene was defined as the sum of the expression of the gene in all cells from that cluster.

---

clusters. y-axis shows the per cent of cells positive for the indicated *TRDV2* (blue) and *TRGV9* (white) gene within each BC γδ T-cell cluster (x-axis). Numbers above bars show the number of positive cells. **(E)** UMAP of BC γδ T cells coloured by the expression of a selected set of markers for identification of IFNG and IL17A subtypes. Grey indicates no expression and purple for high expression. **(F)** Distribution of gene signature scores (y-axis) for IFNγ production, IL17A production, cytotoxicity, adaptive (i.e., antigen presentation on MHC class 1), and innate gene sets in each of the BC γδ T-cell cluster (x-axis). **(G)** Kaplan–Meir survival curve of the The Cancer Genome Atlas breast cancer data (Ciriello et al, 2015). Patients were partitioned into high and low group based on scores for gene signatures of each of the indicated BC γδ T-cell cluster. y-axis shows overall survival. **(H)** Heat map showing top differentially expressed genes (row labels) between the three BC γδ T-cell subtypes. Yellow represents high expression and purple represents low expression.

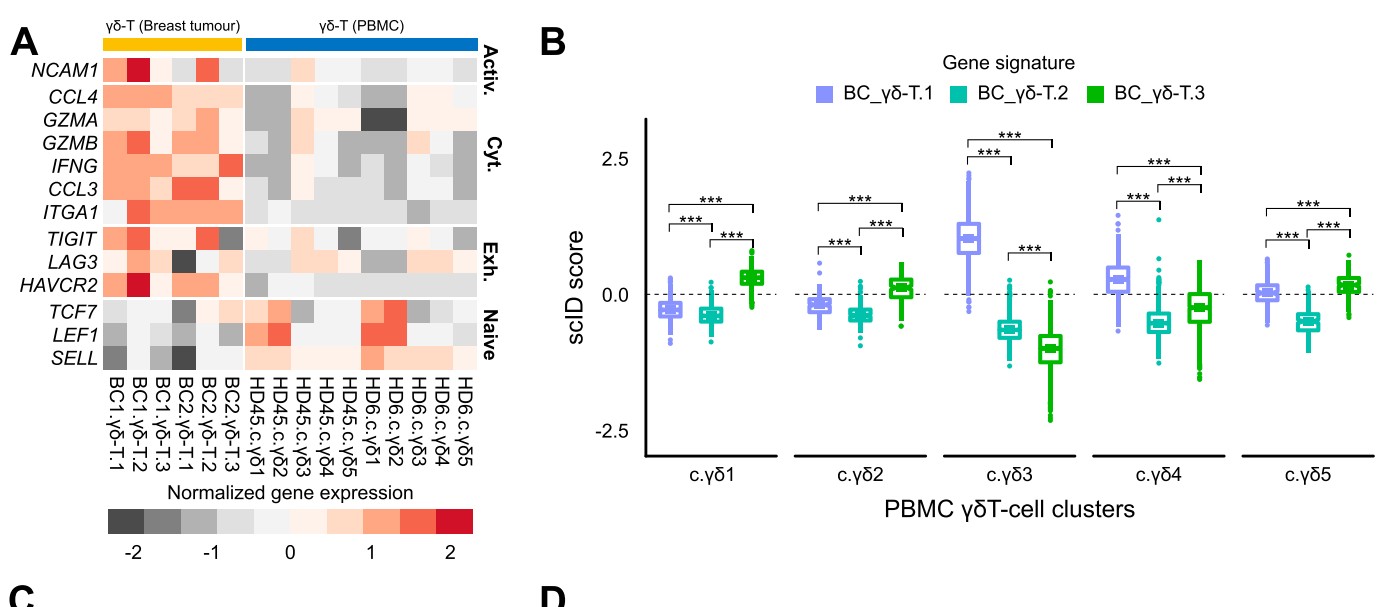

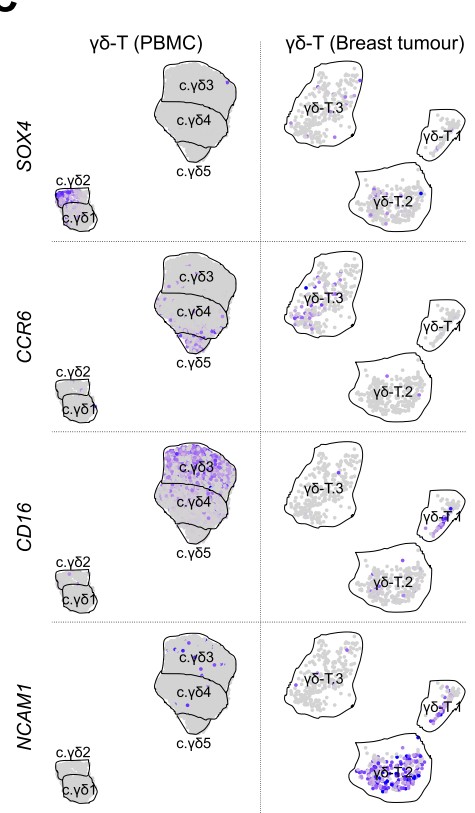

**Figure 4. Comparison of γδ T cells from peripheral blood and breast tumour.**

**(A)** Comparison of expression of selected genes (rows) involved in activation, cytotoxicity, exhaustion, and naive T-cell state between blood γδ T-cell and breast tumour-infiltrating γδ T-cell subtypes. Grey represents low average expression and red represents high average expression of the genes in each subtype (columns). **(B)** Assessment of similarity of the γδ T-cell subtypes in blood and breast tumour. Boxplot of scID scores (y-axis) of the BC cluster-specific gene signatures in the blood clusters (x-axis). Scores above the dashed line indicates enrichment of the indicated gene signature. Mann–Whitney U test was used to compute statistical significance of different scores within each cluster. "***" indicates $P < 0.001$. **(C)** Feature plots showing expression of suggested cluster defining markers in the γδ T-cell subtypes in blood (top row) and BC (bottom row). Grey indicates low expression and purple indicates high expression. **(D)** Table summarizing the proposed refinement of subtype classification of γδ T cells supported by the scRNA-seq data from this study.

### Enrichment of functional gene sets in the γδ T-cell subtypes

We used scID (Boufea et al, 2020) but with equal weights for all genes to calculate an enrichment score per cell for the gene signatures. To identify highly enriched genes we manually selected thresholds based on the distribution of the scores in the cells within each dataset. The lists of genes within the gene signatures were curated from literature and are as follows: *IFNγ* gene signature (*TBX21*, *EOMES*, *STAT1*, *STAT4*, *IL12RB*, and *IFNG*), *IL17* gene signature (*RORC*, *IL23R*, *CCR6*, *IL1R1*, *RORA*, *BLK*, and *IL17A*), cytotoxicity gene signature (*GZMA*, *GZMB*, *GZMK*, *GZMM*, *GZMH*, *PRF1*, *TRAIL*, *FAS*, and *IL12*), antigen presentation gene signature (*HLA-DQPB1*, *HLA-DRA*, and *HLA-DPA1*), and innate gene signature (*KLRK1*/NKG2D and *HCST*/DAP10).

### Cytolytic score

The cytolytic score was calculated for the TCGA breast cancer data (https://portal.gdc.cancer.gov/projects/TCGA-BRCA) as the geometric mean of the expression of *PRF1* and *GZMA* within each sample.

### Survival analysis

We performed survival analysis to test if presence or absence of any γδ T-cell subtype was associated with survival rates of patients with breast cancer. We obtained RNA-seq and clinical data for the TCGA breast cancer study (Ciriello et al, 2015) from Genomic Data Commons Data Portal (https://portal.gdc.cancer.gov/projects/TCGA-BRCA). We filtered the samples based on purity with the following criteria. We removed samples with low purity as they are unlikely to reflect immune cells altered by presence of tumour. We also removed samples with very high purity as they will contain only a small proportion of immune cells that is unlikely to consist of γδ T cells given their rareness. This filtering resulted in 191 samples with purity between 0.6 and 0.7. From the breast tumour single-cell RNA-seq datasets, gene signatures for each of the γδ T cells clusters were extracted using the MAST method as implemented in the Seurat R package, but comparing each γδ T-cell cluster to all other CD45$^+$ clusters to ensure these signatures do not pick up signals from other immune cells in the TCGA RNA-seq samples. We next calculated an enrichment score for each sample and each of the γδ T subtype gene signatures, as the weighted average of the expression of these gene signatures in each TCGA sample, with weights being the log fold change values obtained from differential expression analysis. Then, for each signature, we selected the bottom and top 1/3 of the samples to represent the samples with low and high expression of the gene signature accordingly and performed survival analysis between the two groups using the survfit function of the survival R library (Version 2.44.1.1).

### Mapping across datasets with scID

To identify BC-equivalent γδ T-cell subpopulations in the blood we used scID and calculated a score of each blood cell for each of the BC γδ T-cell subcluster signature. Each signature consists of genes that are up-regulated and genes that are down-regulated in the reference BC cluster, thus positive scID scores indicate cells that express the up-regulated genes and not the down-regulated and negative scores indicate cells that express the down-regulated genes and not the up-regulated.

### Flow cytometry–based validation of clusters

We selected top differentially expressed surface markers for which antibodies were commercially available. Whole blood samples were collected with EDTA 3 healthy donors. Blood samples were purified from whole blood using Ficoll–Paque PLUS according to the manufacturer's protocol. Blood samples were washed with 1× Dulbecco's PBS and cryopreserved in 10% DMSO at −80°C.

Fluorophores were optimised for the panel using the *BioLegend Fluorescence Spectra Analyzer* (BioLegend, 2019. http://www.biolegend.com/spectraanalyzer) to minimise spectral overlap. Antibodies were titrated on healthy donor blood. Blood samples were thawed and rested for 30 min in fresh DMEM supplemented with 10% FBS. Cells were filtered with a 40-μm cell strainer and incubated with *Human TruStain FcX* (BioLegend) for 20 min before staining. The cells were incubated with antibody (Table S1) for 30 min at 4°C and washed twice with FACS buffer (1× PBS, 1% BSA, and 1 mM EDTA) before a final resuspension in FACS buffer. Compensation was performed using *UltraComp eBeads* (Invitrogen) compensation beads incubated with 1 μl of antibody using the same protocol as was used with the blood. Samples were analysed on a *BD LSRFortessa* cell analyser. Flow cytometry files (.fcs) were analysed using *FlowJo* (version 10.0).

## Data Availability

All the code and processed data for generating the figures in this article are available at: https://github.com/BatadaLab/gdT_paper_analysis. Raw sequence data in fastq format and gene counts in matrix market format produced by the cell ranger pipeline (for hg19 assembly) have been deposited in Gene Expression Omnibus with study accession number GSE141665. (1) HD45 (GSM4210786), (2) HD6 (GSM4210787), (3) BC1 (GSM4210788), and (4) BC2 (GSM4210789). Differentially expressed genes are provided in Supplementary Tables.

## Supplementary Information

## Acknowledgements

This study was funded by Chancellor's Fellowship from University of Edinburgh and Wellcome Trust Seed Award (206077/Z/17/Z) to NN Batada. We would like to thank Bruno-Silva Santos (Instituto de Medicina Molecular, Portugal), Dietmar Zaiss (University of Edinburgh, UK) and Matthias Eberl (Cardiff University, UK) for feedback on the earlier version of the manuscript.

## Author Contribution

K Boufea: software, formal analysis, investigation, visualization, methodology, and writing—review and editing.
V González-Huici: investigation.
M Lindberg: validation.
S Symeonides: resources.
O Oikonomidou: resources.
NN Batada: conceptualization, data curation, formal analysis, supervision, funding acquisition, validation, investigation, visualization, methodology, project administration, and writing—original draft, review, and editing.

## Conflict of Interest Statement

The authors declare that they have no conflict of interest.

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
