## [Reviewer comments · Life Science Alliance]

Life Science Alliance

Discovery of a $\gamma\delta$ T-cell subtype associated with favourable outcome of human breast cancer patients

Katerina Boufea, Victor Huici, Marcus Lindberg, Stefan Symeonides, Olga Oikonomidou, and Nizar Batada

DOI: <https://doi.org/10.26508/lsa.202000680>

Corresponding author(s): Nizar Batada, University of Edinburgh

Review Timeline:

Submission Date:	2020-02-18
Editorial Decision:	2020-03-05
Revision Received:	2020-10-05
Editorial Decision:	2020-10-30
Revision Received:	2020-11-06
Accepted:	2020-11-09

Scientific Editor: Shachi Bhatt

Transaction Report:

March 5, 2020

Re: Life Science Alliance manuscript #LSA-2020-00680-T

Dr. Nizar N Batada
Ontario Institute of Cancer Research
Cancer Genomics
101 College Street
Mars south tower, suite 800
toronto, ontario m5g 0a3
Canada

Dear Dr. Batada,

Thank you for submitting your manuscript entitled "Discovery of a $\gamma\delta$ T-cell subtype associated with favourable outcome of human breast cancer patients" to Life Science Alliance. The manuscript was assessed by expert reviewers, whose comments are appended to this letter.

As you will see, the reviewers had somewhat split views on your work. Reviewer #1 does not support further consideration of publication, while reviewer #3 only raises a few concerns that seem addressable. Importantly, however, reviewer #2 notes that the main conclusions drawn need further validation (also pointed out by reviewer #1).

I have discussed your work in light of this input and concluded that we can offer to consider a revised version of your work, should you be able to respond to the reviewers' concerns in a good way and particularly to those raised by reviewer #2. I would be happy to discuss the revision with you further once you have had the time to consider the points carefully. Please let me know in case you would rather prefer to withdraw your manuscript at this stage in light of the reviewer requests to allow submission elsewhere.

Thank you for this interesting contribution to Life Science Alliance. We are looking forward to receiving your revised manuscript.

Sincerely,

B. MANUSCRIPT ORGANIZATION AND FORMATTING:

Reviewer #1 (Comments to the Authors (Required)):

The general aim of the study is to depict scRNASeq profile of human GD T lymphocytes in peripheral blood and in breast cancer. Although dealing with an interesting theme, this report Unfortunately, presents only a mere reproduction of earlier studies. But here, these present a smaller dataset (less cells, less genes per cells, less QC preprocessing), produced by fragile methodologies and without any biological validation. In its current state, this study cannot deserve any publication

The report should answer the following questions:

· what are the major claims and how significant are they?

The major claims of the study are 1{degree sign} identification of the TCRVd1 and TCRVd2 subsets of gd T cells in blood and in breast cancer, 2{degree sign}) the relative functional hallmarks of these respective cell subsets, which presence is associated to breast cancer (BC) outcome.

· are the claims convincing?

No. Results are totally oversold

1{degree sign}) The two subsets (TCRVd1 and TCRVd2) of blood gd T cells are already known and their respective scRNAseq profiles already described with more details, cells, and genes per cell in earlier literature.

2{degree sign}) none of the transcriptome-based observations is substantiated by any wet lab biological assay.

· are the claims appropriately discussed in the context of earlier literature?

No.

· is the study of interest?

No.

There are several major technical and conceptual flaws such as :

-Authors should explain how their 10X genomics 3' chemistry-based dataset, composed of 90 nucleotides-long reads of the 3' side of transcripts allows them to assess the rearranged 5' side-located gamma V gene segments constitutive of the cell surface TCR gamma delta (data Figs 1,3,4). Could such results replace those from the 10X genomics 5' chemistry platform , which is specifically designed for identifying immune repertoires ?

-Authors should specify whether their pre-processing of the data ruled out the HLA genes and sex-related genes prior to the PCA ? These genes are well known to introduce irrelevant but cluster-differentiating genes between donors . Also presumably , all BC patients are females, so were the HD also females ?

-The method applied to isolate the BC biopsy-derived gd T cells excludes neither T CD8 nor NK

cells.

-The paper lacks any validation of the actual specificity of the « BC gd T signatures » for gd T cells versus any other class of cytolytic lymphoid cell (NK1, NK2, ILC3, MAIT, CD8) possibly contributing to anti cancer cytotoxicity.

-The association of the BC gd T signature with survival is very weak from a methodological standpoint.

- Lines 148-149 witness of a lack of knowledge about the basics of TCR gene rearrangements

· are there other experiments that would strengthen the paper?

1{degree sign}) more rigorous methodologies and preprocessing of the data are required throughout the manuscript.

2{degree sign}) the study systematically lacks integration of its scRNAseq data to the already published similar ones which are publicly available.

This encompasses: PBMC from healthy donors, purified GD T cells from healthy donors, global TILs from breast and other cancers. In absence of any such mandatory controls all along the manuscript, the conclusions tentatively drawn by authors are never supported by the data shown. Since such controls are already and publicly available from the GEO dataset repository, this work needs a complete revision to be considered.

3{degree sign}) the study systematically lacks any biological validation of the findings claimed from the scRNASeq analyses. E.g. validate that BC -derived gd TILs are « cytotoxic and immunosuppressed »

4{degree sign}) to validate the specificity of each signature for gd T lymphocytes, versus all other types of cytotoxic immune cells.

For manuscripts that may merit further consideration, it is also helpful if referees can provide advice on the following points where appropriate:

· how to do the study justice without overselling the claims

· There are several errors all along the text, requiring a text revision by a co-author with scientific expertise in human GD T lymphocytes.

To limit this study to characterize more reliably and in full detail the hallmarks of the BC-derived gd T cells, relative to the other cell types from the immune infiltrate in this cancer, and showing biological validation of the corresponding claims

· how to represent earlier literature more fairly

· The text introduces hallmarks of human gd T cells, but (Line 18) this sentence and its associated reference (Ribot et al) are relevant to murine gd T cells, not human ones.

- Line 20 : NKG2D is uniformly expressed by all TCRVdelta subsets of human gd T cells, from and after their central memory stage. CD16 is only expressed at the terminally differentiated stage of all TCRVdelta subsets of human gd T cells. Likewise for all gd T cells, CD28 is a central memory marker defining an intermediate temporal stage of maturation, not a 'different subset' of cells.
- Line 31 : When referring to the publicly available dataset GSE128223 [~8,000 purified GD cells with 1370 genes per cell] as made of 'a relatively low number of cells' (line 31), it is inappropriate to present the present dataset GSE141665 [~6,000 purified GD cells with 1084 genes per cell] as 'a large reference dataset' (line 44).
- Figs 1A,B,C, Fig. 2A,C, Fig 3A-C should be delineated and lack axes and scales.
- Fig. 1B shows no gd T cells from HD 4 and HD 5. This should be clarified.
- how to improve the presentation of methodological detail so that the experiments can be reproduced
- in many points (see above)

Reviewer #2 (Comments to the Authors (Required)):

In this manuscript, Boufea et al. characterized gd T cells in peripheral blood of healthy people and in breast cancer tissues using single cell RNA sequencing. They identify several clusters of cells that may represent functionally distinct subsets in blood and cancer tissues. They propose new surface markers that would enable the characterization/isolation of these subsets by flow cytometry. They identify one subset specifically present in breast cancer tissues compared to blood. Using publicly available data for breast cancer, they propose that this subset may be correlated with positive outcome. The datasets generated are very valuable to analyze the function of gd T cells in cancer, and explore their potential immuno-therapeutic applications. However, the analysis performed is not completely appropriate given the limitations of scRNAseq (eg: dropout). Furthermore, the findings should be confirmed by other approaches such as flow cytometry, to support their functional relevance. Below are my detailed comments:

1. TCR gene usage:

It is likely that the cell clustering is driven by clusters of genes related to effector functions rather than the TCR genes. Even though it is well appreciated that effector functions can be correlated with gdTCR usage, it is not appropriate to assume they are fully correlated. Consistently, although clusters show overrepresented usage of either Vd1 or Vd2 chains, the separation between the two subtypes is not complete (Figure 1C). It is thus not appropriate to identify the clusters as either Vd1 or Vd2 T cells (Figure 1A). In Figure 1D, the representation of the different TCR chains should be showed for both gamma and delta. Also, it is probably more relevant to show the % of each TCRg and d chain within the TCRg+ and TCRd+ cells respectively, since most cells do not express

detectable levels of TCRgenes. Same for figure 3D.

Similarly, there are no complete correlation between clusters and Vg usage. Furthermore, the pairing TCRVg and d is not examined at the single cell level. Therefore, the statement "Thus, we observe three Vδ2Vy9 and two Vδ1Vy4 clusters in blood" should be removed.

Finally, the scRNAseq might not detect all TCR chain equally as observed by the authors, since Vd1 that are the major gd T cell subset in breast cancer are not detected in the samples analyzed. Altogether, these results show that the interpretation regarding TCR usage should be made very carefully, and confirmed by alternative approaches (eg: flow cytometry using TCR specific Ab or PCR for specific TCR chains)

2. Cell clustering

The choice of the 5 clusters seems arbitrary, as there is no clear separation of the subclusters within each macro-cluster based the UMAP or gene expression levels. The different clusters could be driven by cell cycle for example. The scRNAseq analysis it not well explained, so I don't know whether the mitochondrial genes, cell cycle genes, and total number of genes where used for regression.

Consistent with the possible lack of relevance of the subclustering, the interpretation of the different clusters is problematic. For example, the authors state "the δ2.3 cluster is a clear IL17A producer". Although this cluster has a higher score for IL17A production (Figure 1G), T cell cytotoxicity is the top enriched biological process for this cluster (Figure 1 F). Because these two effector functions are generally not co-expressed, this might suggest that the sub-clustering is driven by very few genes, and is not functionally relevant. It might help to show the expression of individual genes related to effector functions in each cluster.

3. Functional relevance

The analysis of GPR56 and CXCR6 expression on Vd2 T cells in blood as a way to distinguish functionally distinct clusters is interesting but insufficient. The function and gene expression of the 3 candidate subsets found in PBMC should be characterized to show they correspond to the clusters identified by scRNAseq and are functionally relevant (eg: qPCR for key genes differentially expressed in scRNAseq, cytokine expression following PMA iono activation). Similarly, the 3 clusters found in breast cancer tissues should be characterized to demonstrate they correspond to functionally distinct subsets. The gdT2 cluster is of particular interest as it is specifically present in breast compared to blood, and its frequency might be correlated with positive prognostic. TCR usage should be characterized, as Vd1 T cells but not Vd2 T cells have been proposed to be correlated with positive prognostic in breast cancer.

Reviewer #3 (Comments to the Authors (Required)):

Boufea and colleagues present a concise study of human peripheral blood-derived and tumor-infiltrating gd T cells using single cell RNA-Seq methods. Specifically, they investigated samples of sorted gd T cells from three adult PBL donors and two samples of unsorted tumor-infiltrating lymphocytes.

Although the RNA-Seq protocol employed (only 90 bp reads from the 3') was not specifically designed to identify the TRG and TRD chains that together build the gd TCR, they were able to identify and further characterize 5 subsets of PBL-derived gd T cells as well as three types of

tumor-infiltrating gd T cells. The gene signature associated to one of the tumor-derived gd T cell subsets (here called BC gd-T2), which was not found as such in the three PBL samples, turned out to be a favorable predictive marker for the overall survival of patients in a large breast cancer cohort published by Ciriello et al., 2015.

Together, this is an important contribution to the field. The data presented adds new information that extends prior scRNA-Seq studies characterizing gd T cell subsets, e.g. Pizzolato, Kaminski et al. PNAS 2019, and nicely complements a recently published study on protective gd T cells in human breast cancer by Wu et al STM 2019.

I only have a few minor points that should be addressed prior to publication:

1. the authors filtered their scRNA-Seq data for TRD positive clusters found in both datasets (see line 58f), which is probably a valid approach. However, some questions arise: Was there one or more TRD positive cluster found only in one of the datasets? If yes, please briefly describe those as they could represent interesting inter-individual variations. Second, it would be helpful to provide actual numbers of the detected TRD, TRG, TRDV1 and TRDV2 positive cells.
2. The authors should critically discuss whether their gd-T2 gene signature is actually specific for gd T cells.
3. Fig. 1B can be misleading as the clusters of the two samples H4/5 and H6 are too near and overlapping, pls. separate clearly.

Reviewer #1

The general aim of the study is to depict scRNASeq profile of human GD T lymphocytes in peripheral blood and in breast cancer. Although dealing with an interesting theme, this report Unfortunately, presents only a mere reproduction of earlier studies. But here, these present a smaller dataset (less cells, less genes per cells, less QC preprocessing), produced by fragile methodologies and without any biological validation. In its current state, this study cannot deserve any publication

The report should answer the following questions:

- what are the major claims and how significant are they?

The major claims of the study are 1{degree sign} identification of the TCRVd1 and TCRVd2 subsets of gd T cells in blood and in breast cancer, 2{degree sign} the relative functional hallmarks of these respective cell subsets, which presence is associated to breast cancer (BC) outcome.

- are the claims convincing?

No. Results are totally oversold

1{degree sign} The two subsets (TCRVd1 and TCRVd2) of blood gd T cells are already known and their respective scRNAseq profiles already described with more details, cells, and genes per cell in earlier literature.

2{degree sign} none of the transcriptome-based observations is substantiated by any wet lab biological assay.

Clearly you have not read the paper. The FACs based validation of markers are shown in Figure 2.

- are the claims appropriately discussed in the context of earlier literature?

No.

- is the study of interest?

No.

There are several major technical and conceptual flaws such as :

-Authors should explain how their 10X genomics 3' chemistry-based dataset, composed of 90 nucleotides-long reads of the 3' side of transcripts allows them to assess the rearranged 5' side-located gamma V gene segments constitutive of the cell surface TCR gamma delta (data Figs 1,3,4). Could such results replace those from the 10X genomics 5' chemistry platform , which is specifically designed for identifying immune repertoires ?

In the 10x's standard protocol, the first read is where the cell barcode (16 bp) and UMI (10bp) is obtained. The second read, which in our case was 91bp for data HD45 and HD6 and 75bp for data BC1 and BC2, will typically read the middle or even the region near the 5' end of the transcript depending on the transcript's length. Fortunately, in our case, this was sufficient to unambiguously identify TRDC, TRDV2 and TRGV9 as outlined below:

- 1) First, we obtained all the VDJ genes sequences from IMGT database
- 2) Next, we aligned all the read 2 for BC1 (which contained were all the CD45+ cells) using the BWA aligner and retained uniquely mapped reads (i.e. `mapq > 60`) only allowing upto 2 mismatches without any gaps
- 3) Lastly, we visualized the distribution of positive cells (using Feature plot overlaying on annotated clusters) and retained those TRD and TRG transcript alignments that were exclusively present in the TRDC+ (i.e. gd T) cell types. Using the above procedure, we retained only *TRDV2* and *TRGV9* gene segments

-Authors should specify whether their pre-processing of the data ruled out the HLA genes and sex-related genes prior to the PCA ? These genes are well known to introduce irrelevant but cluster-differentiating genes between donors . Also presumably , all BC patients are females, so were the HD also females ?

All the HD donors were male. All the BC donors were female. Thus our clustering is not driven by sex specific genes

-The method applied to isolate the BC biopsy-derived gd T cells excludes neither T CD8 nor NK cells.

Only *CD3+* *TRDC+* double positive cells were classified as gd T cells. Conventional CD8 T cells do not express TRDC (a marker also used in Pizzolato et al 2019 paper). NK cells are TRDC- CD3- CD56+. There are many reports showing that a subset of gdT cells can express CD8A (<https://www.jimmunol.org/content/197/12/4584>; <https://www.ncbi.nlm.nih.gov/pmc/articles/PMC4310324/>; https://www.cytometry.org/web/q_view.php?id=284&filter=Interpretation%20and%20Clinical%20Application)

-The paper lacks any validation of the actual specificity of the « BC gd T signatures » for gd T cells versus any other class of cytolytic lymphoid cell (NK1, NK2, ILC3, MAIT, CD8) possibly contributing to anti cancer cytotoxicity.

This question is the same as that raised by reviewer 3. The results are shown in **Supplementary Figure 2**. To address this question, we scored all the CD45 positive clusters in the breast cancer data (including CD4, CD8, NK, Monocytes and gdT) and computed their score for each of gdT cluster signatures (**Supplementary Figure 2**). Only the respective gd-T cluster had a significantly higher score for that cluster's gene signature. Notably the ab-T cells such CD4 and CD8 (which are also known to be associated with better survival in many cancers) did not score high with these signatures. This result

shows that the scores of gene signature in TCGA BRCA data that we have used for survival analysis are very specific to gd-T subtypes identified in our BC scRNA-seq dataset.

-The association of the BC gd T signature with survival is very weak from a methodological standpoint.

It's unclear how the author defines weak or whether they mean weak biologically or statistically. We have used the standard approach used for survival analysis (ie. a Log Rank P-value of the Kaplan Meier curve) which is statistically significant

- Lines 148-149 witness of a lack of knowledge about the basics of TCR gene rearrangements

· are there other experiments that would strengthen the paper?

1{degree sign}) more rigorous methodologies and preprocessing of the data are required throughout the manuscript.

The single cell expression data presented in our manuscript were filtered based on stringent criteria such as removal of cells that had an excessive number of mitochondrial genes or had low number of detected genes or low ratio of housekeeping genes to the total number of gene expressed. Moreover, we used a published cell cycle classification tool (in the R package scran) to compute the proportion of G1/G2/S phase genes in each cluster. The proportions are not significantly different between the clusters) (**Supplementary Figure 1A**) and reclustering the data after discarding all cell cycle and mitochondrial genes, recovered the same clusters (**Supplementary Figure 1B**). We therefore conclude that the clusters are not driven by cell cycle genes or obvious artefacts.

The presence of shared boundaries between subclusters in the UMAP space do not necessarily mean that clusters are wrong, not least because dimension reduction (i.e. UMAP) is done for visualization while cells are clustered in a higher dimensional space where the clusters may be well separated. For example, it is well appreciated that even though subsets of CD4 and CD8 positive T cells are functionally quite distinct, single cell RNA-seq clusters of these subsets do not separate out (**Figure 3A**) during dimension reduction via TSNE or UMAP despite clearly express lineage specific markers (CD4 and CD8 in this case). This can also be seen in the Pizzolato et al 2019 paper (Figure 1C). Nevertheless, we have performed additional analysis to address the concern raised by the reviewer regarding clustering by subsampling of the data and reclustering which revealed that the cluster number and cluster memberships are stable (see **Supplementary Figure 1C**).

The clusters were identified unbiasedly as outlined in the methods section and supported by multiple biologically meaningful features: a) mutually exclusive groups of cells expressing CCR6 and RORC (marker of IL17A producers) and IFNG (marker of cytotoxic gdT cells). b) mutual exclusive expression of two published markers (i.e. CD16 and CD28) defining two subtypes of TRDV2+ cluster (Figure 2), c) the pre-merged individual datasets (HD45 and HD6) had exactly the same number of clusters

We hope that the reviewer is satisfied with our additional analysis and agrees that the clusters are not arbitrary nor reflect artifacts from mitochondrial or cell cycle but instead are biologically meaningful.

2{degree sign}) the study systematically lacks integration of its scRNAseq data to the already published similar ones which are publicly available.

This encompasses: PBMC from healthy donors, purified GD T cells from healthy donors, global TILs from breast and other cancers. In absence of any such mandatory controls all along the manuscript, the conclusions tentatively drawn by authors are never supported by the data shown. Since such controls are already and publicly available from the GEO dataset repository, this work needs a complete revision to be considered.

Our BC scRNA-seq data had all the CD45+ (TILs) from breast cancer and have presented it in Figure 3A.

3{degree sign}) the study systematically lacks any biological validation of the findings claimed from the scRNASeq analyses. E.g. validate that BC -derived gd TILs are « cytotoxic and immunosuppressed »

We have shown this via expression of gene of various classes of genes defined as cytotoxic and immunosuppressed as done in Tirosh et al 2016 paper (<https://pubmed.ncbi.nlm.nih.gov/27124452/>). We are happy to remove this statement from the manuscript if it pleases the reviewer as this point is very minor and does not affect the message of the paper.

4{degree sign}) to validate the specificity of each signature for gd T lymphocytes, versus all other types of cytotoxic immune cells.

To address this question, we scored all the CD45 positive clusters in the breast cancer data (including CD4, CD8, NK, Monocytes and gdT) and computed their score for each of gdT cluster signatures (**Supplementary Figure 2**). Only the respective gd-T cluster had a significantly higher score for that cluster's gene signature. Notably the ab-T cells such CD4 and CD8 (which are also known to be associated with better survival in many cancers) did not score high with these signatures. This result shows that the scores of gene signature in TCGA BRCA data that we have used for survival analysis are very specific to gd-T subtypes identified in our BC scRNA-seq dataset.

For manuscripts that may merit further consideration, it is also helpful if referees can provide advice on the following points where appropriate:

- how to do the study justice without overselling the claims
- There are several errors all along the text, requiring a text revision by a co-author with scientific expertise in human GD T lymphocytes.

To limit this study to characterize more reliably and in full detail the hallmarks of the BC-derived gd T cells, relative to the other cell types from the immune infiltrate in this cancer, and showing biological validation of the corresponding claims

- how to represent earlier literature more fairly

- The text introduces hallmarks of human gd T cells, but (Line 18) this sentence and its associated reference (Ribot et al) are relevant to murine gd T cells, not human ones.

We have now updated the sentence to point out the result is from mouse studies

- Line 20 : NKG2D is uniformly expressed by all TCRVdelta subsets of human gd T cells, from and after their central memory stage. CD16 is only expressed at the terminally differentiated stage of all TCRVdelta subsets of human gd T cells. Likewise for all gd T cells, CD28 is a central memory marker defining an intermediate temporal stage of maturation, not a 'different subset' of cells.

We have removed the sentence regarding NKG2D (line 20). CD16 and CD28 were defined as mutually exclusive markers of subtypes within the TRDV2+ human peripheral blood gd-T cells by Ryan et al 2016 (<https://www.pnas.org/content/113/50/14378>)

- Line 31 : When referring to the publicly available dataset GSE128223 [~8,000 purified GD cells with 1370 genes per cell] as made of 'a relatively low number of cells' (line 31), it is inappropriate to present the present dataset GSE141665 [~6,000 purified GD cells with 1084 genes per cell] as 'a large reference dataset' (line 44).

The reviewer is referring to the PNAS paper by Pizzolato et al 2019 which we have discussed in the introduction. In this paper, cells were sorted with anti-TCRVdelta1 and anti-TCRVdelta2 antibodies which were then sequenced separately. No subclusters were defined in this paper. In contrast, we used a pan-TCR gd antibody for isolating gd-T from blood and anti-CD45+ and computational analysis for identifying gd-T cells from breast tumours which is a tissue not represented in the Pizzolato et al paper.

- Figs 1A,B,C, Fig. 2A,C, Fig 3A-C should be delineated and lack axes and scales.

These figures are UMAP (a non-linear dimension reduction method) 2 dimensional projections and their scale do not obviously provide biological meaning. We have left them out (as done in many other single cell RNA-seq papers) to reduce clutter in the figures.

- Fig. 1B shows no gd T cells from HD 4 and HD 5. This should be clarified.

The cells labelled HD45 are a merge of HD4 and HD5 individuals. These two samples were not barcoded (and as we did have the genotype data) we did not have a way to properly segregate them without causing bias.

- how to improve the presentation of methodological detail so that the experiments can be reproduced

· in many points (see above)

Reviewer #2

In this manuscript, Boufeia et al. characterized gd T cells in peripheral blood of healthy people and in breast cancer tissues using single cell RNA sequencing. They identify several clusters of cells that may represent functionally distinct subsets in blood and cancer tissues. They propose new surface markers that would enable the characterization/isolation of these subsets by flow cytometry. They identify one subset specifically present in breast cancer tissues compared to blood. Using publicly available data for breast cancer, they propose that this subset may be correlated with positive outcome. The datasets generated are very valuable to analyze the function of gd T cells in cancer, and explore their potential immuno-therapeutic applications. However, the analysis performed is not completely appropriate given the limitations of scRNAseq (eg: dropout). Furthermore, the findings should be confirmed by other approaches such as flow cytometry, to support their functional relevance.

We thank the reviewer for appreciating the value of our data and address the questions raised below

Below are my detailed comments:

1. TCR gene usage:

(2.1) “It is thus not appropriate to identify the clusters as either Vd1 or Vd2 T cells”

It is likely that the cell clustering is driven by clusters of genes related to effector functions rather than the TCR genes. Even though it is well appreciated that effector functions can be correlated with gdTCR usage, it is not appropriate to assume they are fully correlated. Consistently, although clusters show overrepresented usage of either Vd1 or Vd2 chains, the separation between the two subtypes is not complete (Figure 1C). It is thus not appropriate to identify the clusters as either Vd1 or Vd2 T cells (Figure 1A).

We agree with the reviewer's suggestion that the clusters appear to be driven more by effector function rather than TCR. As per reviewer's suggestion, instead of referring to these clusters as TRDV1 and TRDV2 clusters, in the updated manuscript, we have labelled these as c.gd1, c.gd2, ..., c.gd5.

(2.2) Show the % of each TCRg and d chain within the TCRg+ and TCRd+ cells respectively

In Figure 1D, the representation of the different TCR chains should be shown for both gamma and delta. Also, it is probably more relevant to show the % of each TCRg and d chain within the TCRg+ and TCRd+ cells respectively, since most cells do not express detectable levels of TCRgenes. Same for figure 3D.

We have now created the figure as requested by the reviewer. The updated figures in the manuscript are Figure 1D and Figure 3D.

(2.3) The statement "Thus, we observe three Vδ2Vγ9 and two Vδ1Vγ4 clusters in blood" should be removed.

Similarly, there is no complete correlation between clusters and Vg usage. Furthermore, the pairing TCRVg and d is not examined at the single cell level. Therefore, the statement "Thus, we observe three Vδ2Vγ9 and two Vδ1Vγ4 clusters in blood" should be removed.

We agree with the reviewer. This sentence no longer appears in the updated manuscript.

(2.4) The interpretation regarding TCR usage should be made very carefully

Finally, the scRNAseq might not detect all TCR chain equally as observed by the authors, since Vd1 that are the major gd T cell subset in breast cancer are not detected in the samples analyzed. Altogether, these results show that the interpretation regarding TCR usage should be made very carefully, and confirmed by alternative approaches (eg: flow cytometry using TCR specific Ab or PCR for specific TCR chains)

We thank the reviewer for point this out. We agree with the reviewer's point and in the updated manuscript we have deleted and rephrased the sentences regarding TCR usage.

2. Cell clustering

(2.5) The choice of the 5 clusters seems arbitrary

The choice of the 5 clusters seems arbitrary, as there is no clear separation of the subclusters within each macro-cluster based the UMAP or gene expression levels. The different clusters could be driven by cell cycle for example. The scRNAseq analysis it not well explained, so I don't know whether the mitochondrial genes, cell cycle genes, and total number of genes where used for regression.

We apologize for not presenting the technical aspects of clustering analysis in more detail. In the updated manuscript and below, we have provided more information.

The single cell expression data presented in our manuscript were filtered based on stringent criteria such as removal of cells that had an excessive number of mitochondrial genes or had low number of detected genes or low ratio of housekeeping genes to the total number of gene expressed. Moreover, we used a published cell cycle classification tool (in the R package scran) to compute the proportion of G1/G2/S phase genes in each cluster. The proportions are not significantly different between the clusters) (**Supplementary Figure 1A**) and reclustering the data after discarding all cell cycle and mitochondrial genes, recovered the same clusters (**Supplementary Figure 1B**). We therefore conclude that the clusters are not driven by cell cycle genes or obvious artefacts.

The presence of shared boundaries between subclusters in the UMAP space do not necessarily mean that clusters are wrong, not least because dimension reduction (i.e. UMAP) is done for visualization while cells are clustered in a higher dimensional space where the clusters may be well separated. For example, it is well appreciated that even though subsets of CD4 and CD8 positive T cells are

functionally quite distinct, single cell RNA-seq clusters of these subsets do not separate out (**Figure 3A**) during dimension reduction via TSNE or UMAP despite clearly express lineage specific markers (CD4 and CD8 in this case). This can also be seen in the Pizzolato et al 2019 paper (Figure 1C). Nevertheless, we have performed additional analysis to address the concern raised by the reviewer regarding clustering by subsampling of the data and reclustering which revealed that the cluster number and cluster memberships are stable (see **Supplementary Figure 1C**).

The clusters were identified unbiasedly as outlined in the methods section and supported by multiple biologically meaningful features: a) mutually exclusive groups of cells expressing CCR6 and RORC (marker of IL17A producers) and IFNG (marker of cytotoxic gdT cells). b) mutual exclusive expression of two published markers (i.e. CD16 and CD28) defining two subtypes of TRDV2+ cluster (Figure 2), c) the pre-merged individual datasets (HD45 and HD6) had exactly the same number of clusters

We hope that the reviewer is satisfied with our additional analysis and agrees that the clusters are not arbitrary nor reflect artifacts from mitochondrial or cell cycle but instead are biologically meaningful.

(2.6) Conflicting annotations of cytotoxicity and IL17 production suggest incorrect clustering

Consistent with the possible lack of relevance of the subclustering, the interpretation of the different clusters is problematic. For example, the authors state "the δ 2.3 cluster is a clear IL17A producer". Although this cluster has a higher score for IL17A production (Figure 1G), T cell cytotoxicity is the top enriched biological process for this cluster (Figure 1 F). Because these two effector functions are generally not co-expressed, this might suggest that the sub-clustering is driven by very few genes, and is not functionally relevant. It might help to show the expression of individual genes related to effector functions in each cluster.

Thank you for catching this contradictory result. We have determined that there is an issue with the gene set enrichment analysis approach that we used -- usually one uses a list of highly differentially expressed genes to compute enrichment relative to a background (i.e. all the genes expressed in that cell type). However, for scRNA-seq data of similar cell types (as is in our case), this is quite challenging because the differences between the subtypes is more subtle and obtaining a large enough list requires using relaxed fold change threshold but also another parameter called percent of cells expressing these genes. The latter is particularly difficult for scRNA-seq due to shallow coverage and dropout which makes it difficult to distinguish truly absent genes from those that were not detected perhaps due to moderate to low expression or other bias. We have therefore decided to remove this analysis from the updated manuscript. However, we trust that our classification based on supervised scoring (i.e. using well known genes for effector functions) is indeed correct as the cluster that is classified as IL17A producer specifically expresses well known markers of that cluster such as RORC and CCR6. We have also included expression of individual marker genes for defining clusters (**Figure 1E**; **Figure 3E**) as asked by the reviewer.

3. Functional relevance

(2.6) The function and gene expression of the 3 candidate subsets found in PBMC should be characterized

The analysis of GPR56 and CXCR6 expression on Vd2 T cells in blood as a way to distinguish functionally distinct clusters is interesting but insufficient. The function and gene expression of the 3 candidate subsets found in PBMC should be characterized to show they correspond to the clusters identified by scRNAseq and are functionally relevant (eg: qPCR for key genes differentially expressed in scRNAseq, cytokine expression following PMA iono activation). Similarly, the 3 clusters found in breast cancer tissues should be characterized to demonstrate they correspond to functionally distinct subsets. The gdT2 cluster is of particular interest as it is specifically present in breast compared to blood, and its frequency might be correlated with positive prognostic. TCR usage should be characterized as Vd1 T cells but not Vd2 T cells have been proposed to be correlated with positive prognostic in breast cancer.

We agree that the followup experiments suggested by the reviewer to independently to further validate the findings from our single cell RNA-seq data would strengthen this study however our computational analysis already presented is rigorous and our FACS analysis shown in Figure 2 validate the mutual exclusion of the two clusters of the TRDV2+ macrocluster. These additional experimental work are difficult to carry out during this time due to extenuating circumstances (COVID associated closures) but we also feel that what we have presented constitutes a strong body of work as it delineates previously unappreciated subtypes in gdT cells, their gene signatures and markers as well as identification of a gdT subtype with clinical relevance.

Reviewer #3

Boufeva and colleagues present a concise study of human peripheral blood-derived and tumor-infiltrating gd T cells using single cell RNA-Seq methods. Specifically, they investigated samples of sorted gd T cells from three adult PBL donors and two samples of unsorted tumor-infiltrating lymphocytes.

Although the RNA-Seq protocol employed (only 90 bp reads from the 3') was not specifically designed to identify the TRG and TRD chains that together build the gd TCR, they were able to identify and further characterize 5 subsets of PBL-derived gd T cells as well as three types of tumor-infiltrating gd T cells. The gene signature associated to one of the tumor-derived gd T cell subsets (here called BC gd-T2), which was not found as such in the three PBL samples, turned out to be a favorable predictive marker for the overall survival of patients in a large breast cancer cohort published by Ciriello et al., 2015.

Together, this is an important contribution to the field. The data presented adds new information that extends prior scRNA-Seq studies characterizing gd T cell subsets, e.g. Pizzolato, Kaminski et al. PNAS 2019, and nicely complements a recently published study on protective gd T cells in human breast cancer by Wu et al STM 2019.

We thank the review for the nice summary and appreciation of the value of the data and analysis presented in our manuscript

I only have a few minor points that should be addressed prior to publication:

1. the authors filtered their scRNA-Seq data for TRD positive clusters found in both datasets (see line 58f), which is probably a valid approach. However, some questions arise:

(3.1)

Was there one or more TRD positive cluster found only in one of the datasets? If yes, please briefly describe those as they could represent interesting inter-individual variations.

As shown in Figure 1B and Figure 3B, we observed equivalent number of TRDC+ gdT clusters in both the blood data (HD45 and HD6) and also found equivalent number of TRDC+ gdT clusters in the both the breast tumour infiltrating immune cell data as well (BC1 and BC2)

(3.2)

Second, it would be helpful to provide actual numbers of the detected TRD, TRG, TRDV1 and TRDV2 positive cells.

We have now plotted the actual number and proportion of detected TRDV and TRDG genes (**Figure 1D and Figure 3D**)

(3.3)

2. The authors should critically discuss whether their gd-T2 gene signature is actually specific for gd T cells.

To address this question, we scored all the CD45 positive clusters in the breast cancer data (including CD4, CD8, NK, Monocytes and gdT) and computed their score for each of gdT cluster signatures (**Supplementary Figure 2**). Only the respective gd-T cluster had a significantly higher score for that cluster's gene signature. Notably the ab-T cells such CD4 and CD8 (which are also known to be associated with better survival in many cancers) did not score high with these signatures. This result shows that the scores of gene signature in TCGA BRCA data that we have used for survival analysis are very specific to gd-T subtypes identified in our BC scRNA-seq dataset.

(3.4)

3. Fig. 1B can be misleading as the clusters of the two samples H4/5 and H6 are too near and overlapping, pls. separate clearly.

We have now fixed the distance between the subplots.

October 30, 2020

RE: Life Science Alliance Manuscript #LSA-2020-00680-TR

Dr. Nizar N Batada
Institute of Genetics & Molecular Medicine
Center for Genetic and Experimental Medicine
Crewe Road S
Edinburgh EH4 2XU
United Kingdom

Dear Dr. Batada,

Thank you for submitting your revised manuscript entitled "Discovery of a $\gamma\delta$ T-cell subtype associated with favourable outcome of human breast cancer patients". We would be happy to publish your paper in Life Science Alliance (LSA) pending final revisions necessary to meet our formatting guidelines.

Along with the points listed below, please also attend to the following:

- please make sure the author order in our system and in the manuscript text match
- please add ORCID ID for corresponding author-you should have received instructions on how to do so
- please use the [10 author names, et al.] format in your references (i.e. limit the author names to the first 10)
- please add the Author Contributions to the main manuscript text
- please add a separate section with your figure legends for both your main and supplementary figures
- please specify the manuscript category when submitting the revised version

A. FINAL FILES:

-- High-resolution figure, supplementary figure and video files uploaded as individual files: See our

detailed guidelines for preparing your production-ready images, <https://www.life-science-alliance.org/authors>

B. MANUSCRIPT ORGANIZATION AND FORMATTING:

Sincerely,

Shachi Bhatt, Ph.D.
Executive Editor
Life Science Alliance
<https://www.lsjournal.org/>
Tweet @SciBhatt @LSAJournal

Reviewer #2 (Comments to the Authors (Required)):

the revised version provided by the authors answered my concerns. The paper is to my view suitable for publication.

Reviewer #3 (Comments to the Authors (Required)):

I think the ms is ok for publication in LSA now. The data are largely in line with and complement the still sparse literature using scRNA-Seq for characterization of tumor-reactive gd T cells.

November 9, 2020

RE: Life Science Alliance Manuscript #LSA-2020-00680-TRR

Dr. Nizar N Batada
University of Edinburgh
Institute of Genetics & Molecular Medicine
Crewe Road S
Edinburgh EH4 2XU
United Kingdom

Dear Dr. Batada,

Thank you for submitting your not used entitled "Discovery of a $\gamma\delta$ T-cell subtype associated with favourable outcome of human breast cancer patients". It is a pleasure to let you know that your manuscript is now accepted for publication in Life Science Alliance. Congratulations on this interesting work.

DISTRIBUTION OF MATERIALS:

Again, congratulations on a very nice paper. I hope you found the review process to be constructive and are pleased with how the manuscript was handled editorially. We look forward to future exciting submissions from your lab.

Sincerely,

Shachi Bhatt, Ph.D.

Executive Editor

Life Science Alliance

<https://www.lsjournal.org/>
